# Performance Analysis of Cardioid and Omnidirectional Microphones in Spherical Sector Arrays for Coherent Source Localization

**DOI:** 10.3390/s24237572

**Published:** 2024-11-27

**Authors:** Chibuzo Joseph Nnonyelu, Meng Jiang, Marianthi Adamopoulou, Jan Lundgren

**Affiliations:** Department of Computer and Electrical Engineering, Mid Sweden University, 851 70 Sundsvall, Sweden; meng.jiang@miun.se (M.J.); marianthi.adamopoulou@miun.se (M.A.); jan.lundgren@miun.se (J.L.)

**Keywords:** hemispherical array, cardioid microphones, spherical sector harmonics, coherent sources, wideband sources

## Abstract

Traditional spherical sector microphone arrays using omnidirectional microphones face limitations in modal strength and spatial resolution, especially within spherical sector configurations. This study aims to enhance array performance by developing a spherical sector array employing first-order cardioid microphones. A model based on spherical sector harmonic (SSH) functions is introduced to extend the benefits of spherical harmonics to sector arrays. Modal strength analysis demonstrates that cardioid microphones in open spherical sectors enhance nonzero-order strengths and eliminate the nulls associated with spherical Bessel functions. We find that the spatial resolution of spherical cap arrays depends on the array’s maximum order and the limiting polar angle, but is independent of the microphone gain pattern. We assess direction-of-arrival (DOA) estimation performance for coherent wideband sources using the array manifold interpolation method, and compare cardioid and omnidirectional arrays through simulations in both open and rigid hemispherical configurations. The results indicate that cardioid arrays outperform omnidirectional ones on DOA estimation tasks, with performance improving alongside increased microphone directivity in the open hemispherical configuration. Specifically, hypercardioid microphones yielded the best results in the open configuration, while subcardioid microphones (without nulls) were optimal in rigid configurations. These findings demonstrate that spherical sector arrays of first-order cardioid microphones offer improved modal strength and DOA estimation capabilities over traditional omnidirectional arrays, providing significantly enhancing performance in spherical sector array processing.

## 1. Introduction

In direction-of-arrival (DOA) estimation, the directions from which sound sources originate are typically determined using measurements obtained by sampling the sound field at multiple closely-spaced locations. DOA has been applied in various fields and systems, including unmanned aerial vehicles and robotics [1,2,3], speaker localization [4,5], fault diagnosis via noise source localization [6,7,8], ground vehicle tracking [9], acoustic cameras for modeling the geometry of the surrounding space [10], and situational awareness for military ground forces [11]. Arrangement of the microphones can take various geometrical configurations based on the specific requirements and performance goals of the system [12] (Chapter 4). A spherical microphone array (SMA) consists of a sphere with microphones strategically positioned on its surface. This spatial configuration ideally provides a three-dimensional symmetric arrangement [13,14]. The continued interest in spherical microphone arrays is due to their ability to capture a three-dimensional sound field, which simplifies processing in the spherical harmonic domain [15,16]. Driven by sustained interest in this technology, various techniques and studies have been developed for SMAs. These include spherical harmonic smoothing [13], modal smoothing [17], DOA sectorization using machine learning methods [18], eigenbeam processing [19], model-based methods [20] performance analysis using Cramer–Rao bounds [21].

Despite the advantages of SMAs, they can be cumbersome, and are often redundant for applications where the sound sources are confined to a known sector of the sphere. To address this issue, the hemispherical harmonics have been used for processing hemispherical arrays, mitigating the discontinuity problem in spherical harmonics on a hemisphere [22]. This approach was later generalized to any spherical sector microphone array (SSMA), leading to the development of spherical sector harmonics (SSH) basis functions [23], which extend the benefits of spherical harmonics to spherical sectors. It has been demonstrated that spherical sector harmonics retain the orthogonality and completeness properties of spherical harmonics [24]. The orthogonality of SSH permits the use of sampling techniques designed for spherical arrays [16] (Chapter 3) in the sampling of the spherical sector with slight adjustment by a scaling factor. The sound source localization and beamforming applications of spherical sector harmonics were investigated in [22,23,24,25], demonstrating significant potential for beamforming and localization using spherical sector arrays. These studies are all based on the spherical sector array of omnidirectional microphones. To the best of our knowledge, no study has yet been carried out on the spherical sector array of cardioid microphones.

First-order cardioid microphones form a family of directional microphones with a gain pattern described by α+(1−α)cos(γ), where γ∈[0,2π) represents the angle of incidence of the incoming sound relative to the microphone’s main axis and α∈(0,1) is the shape parameter (also called the *cardioidicity* index) [26] (Chapter 5). Compared to figure-8 and omnidirectional microphone layouts, cardioid microphones offer improved random efficiency, extended distance factor, and enhanced directivity. First-order cardioid microphones are commercially available with different α values, each optimizing specific microphone properties. With α=0, the shape degenerates to that of a figure-8 microphone layout, while with α=1 it degenerates to an omnidirectional layout. The hypercardioid (α=1/4) maximizes the random efficiency in the forward direction. The supercardioid ($\alpha =\frac{1}{2}\left(\sqrt{3}-1\right)\approx 0.37$) maximizes the front-to-back gain ratio. The standard cardioid (α=1/2) completely suppresses incident sound from the rear of the microphone. The subcardioid (α=0.7) maximizes the beamwidth of a first-order cardioid microphone. The performance of first-order cardioid microphones has been studied previously for a collocated pair [27], collocated diad/triad [28,29,30,31,32], and phased array [33]. These studies demonstrate the advantages of cardioid microphones in beamforming and DOA estimation. The SMA of cardioid sensors has been shown to offer potential performance improvements due to the enhanced robustness to noise achieved by high modal strengths at low modes compared to omnidirectional sensors in similar configurations. Furthermore, the use of cardioid sensors on an open sphere eliminates the zeros of the spherical Bessel function, just as in a rigid sphere array of omnidirectional sensors ([16] (Section 4.3), [34,35]). However, to date this improvement has only been explored for open sphere arrays [16]. To the best of our knowledge, all previous studies on spherical sector arrays have focused exclusively on arrays of omnidirectional microphones. This highlights the need to address a gap in the literature regarding spherical sector arrays using cardioid microphones, especially given the great potentials of cardioid microphones.

This work develops a spherical sector representation of an incident wave sampled by a spherical sector array of cardioid microphones. For DOA estimation of coherent wideband sources, we propose a focusing matrix based on spherical sector harmonics. This matrix allows the constituent narrowband covariance matrices to be steered into a single covariance matrix, which is a full-rank matrix suitable for subspace methods or matrix inversion. The design of this focusing matrix is inspired by the array manifold interpolation method proposed for uniform circular arrays [36,37]. The performance of the spherical sector array using cardioid microphones is compared to that of omnidirectional microphones, with the aim of understanding how the use of directional microphones can enhance the direction-finding performance of spherical sector arrays.

The rest of this paper is organized as follows: in Section 2, the spherical sector harmonics representation of sound pressure is introduced, extended to the first-order cardioid microphone, and studied in terms of mode strength and spatial resolution; in Section 3, the array manifold interpolation method for wideband sources based on the spherical sector harmonics is developed; Section 4 compares the direction-of-arrival estimation performance of the arrays; finally, the work is concluded in Section 5.

## 2. Spherical Sector Harmonics (SSH) Representation

The spherical sector harmonics for a spherical sector array of omnidirectional microphones has been previously studied by Kumari et al. [24]. An overview of SSH for omnidirectional microphone array is provided in Section 2.1, with SSH then extended to a spherical sector array of cardioid sensors in Section 2.2. In addition, we study the mode strengths of the cardioid microphone array in Section 2.3 and their spatial resolution in Section 2.4.

### 2.1. SSH for Omnidirectional Sensors

Similar to the spherical harmonics [13,16,38], the sound pressure can be sampled over a spherical sector using the spherical sector harmonics basis function. Specifically, for an incident plane wave impinging on a spherical sector (see Figure 1 for a hemispherical microphone array) from direction $\Omega_{s}=\left(\theta_{s},\phi_{s}\right)$, the pressure field sampled at a location on the spherical sector $\Psi_{𝓁}=\left({\bar{\theta }}_{𝓁},{\bar{\phi }}_{𝓁}\right)$ is provided as follows [22,23,24]:
(1)$$\begin{array}{ccc} p_{𝓁}\left(k,\Omega_{s}\right) & = & x\left(k\right)e^{ik_{s}^{T}r_{𝓁}}\\ & \approx & \sum_{n=0}^{N}\sum_{m=-n}^{n}p_{n}^{m}\left(kr\right)G_{n}^{m}\left(\Psi_{𝓁}\right)x\left(k\right),\\ & = & \sum_{n=0}^{N}\sum_{m=-n}^{n}b_{n}\left(kr\right){\left[G_{n}^{m}\left(\Omega_{s}\right)\right]}^{*}G_{n}^{m}\left(\Psi_{𝓁}\right)x\left(k\right), \end{array}$$
where $p_{n}^{m}\left(kr\right)=b_{n}\left(kr\right){\left[G_{n}^{m}\left(\Omega_{s}\right)\right]}^{*}$ is the spherical-sector Fourier transform of the incident pressure field, x(k) is the amplitude of the plane wave, *N* is the highest order of the spherical-sector harmonics, $\Omega_{s}=\left(\theta_{s},\phi_{s}\right)$ represents the polar and azimuthal angles (direction of arrival) of the source, $\Psi_{𝓁}=\left({\bar{\theta }}_{𝓁},{\bar{\phi }}_{𝓁}\right)$ represents the polar and azimuthal angles of the *ℓ*-th microphone, and the superscript ^∗^ denotes complex conjugation. The SSH basis function $G_{n}^{m}\left(\cdot \right)$ of degree *n* and order *m* (both integers) for a spherical sector bounded by $\theta \in [\theta_{1},\theta_{2}]$ and ϕ∈[0,2π/u], u≥1 is provided for 0≤n≤N as follows [24]:(2)$$\begin{array}{ccc} G_{n}^{m}\left(\Omega \right) & = & \left\{\begin{array}{cc} K_{n}^{m}P_{n}^{m}\left(q_{1}\cos\theta +q_{2}\right)e^{imu\phi } & 0\leq m\leq n\\ {\left(-1\right)}^{\left|m\right|}G_{n}^{|m|*}\left(\Omega \right) & -n\leq m<0 \end{array}\right. \end{array}$$
where the normalization constant is
(3)$$\begin{array}{ccc} K_{n}^{m} & := & \sqrt{\frac{\left(2n+1\right)\left(n-m\right)!\left(q_{1}u\right)}{4\pi (n+m)!}}, \end{array}$$
and $P_{n}^{m}\left(\cdot \right)$ is the associated Legendre polynomial (ALP) of degree *n* and order *m*. The ALP shifting coefficients $q_{1}\neq 0$ and $q_{2}$ are selected for the bounding polar angles $\theta_{1}$ and $\theta_{2}$ [24]: (4)$$\begin{array}{ccc} q_{1}\cos\left(\theta_{2}\right)+q_{2} & = & -1, \end{array}$$(5)$$\begin{array}{ccc} q_{1}\cos\left(\theta_{1}\right)+q_{2} & = & 1. \end{array}$$

The sectoral mode strength $b_{n}\left(kr\right)$ is provided as follows [23,24]:(6)$$\begin{array}{ccc} b_{n}\left(kr\right) & := & \left\{\begin{array}{cc} w_{n}j_{n}\left(kr\right) & \;\mathrm{for}\;\mathrm{open}\;\mathrm{sector}\\ w_{n}\left[j_{n}\left(kr\right)-\frac{j_{n}^{\prime }\left(kr\right)}{h_{n}^{{\left(2\right)}^{\prime }}\left(kr\right)}h_{n}^{\left(2\right)}\left(kr\right)\right] & \;\mathrm{for}\;\mathrm{rigid}\;\mathrm{sector} \end{array}\right. \end{array}$$
where $j_{n}\left(\cdot \right)$ and $h_{n}^{\left(2\right)}\left(\cdot \right)$ are the spherical Bessel function and spherical Hankel function of second kind with order *n*, respectively, and $j_{n}^{\prime }\left(\cdot \right)$ and $h_{n}^{{\left(2\right)}^{\prime }}\left(\cdot \right)$ are their respective derivatives with respect to their arguments.

The source can either lie in-sector, i.e., $\theta_{s}\in \left[\theta_{1},\theta_{2}\right]$ and $\psi_{s}\in \left[0,2\pi /u\right]$, or anywhere in 3D space, i.e., $\theta_{s}\in \left[0,\pi \right]$ and $\phi_{s}\in \left[0,2\pi \right)$. For these cases, we have the function [23,24]
(7)$$\begin{array}{ccc} w_{n} & := & \left\{\begin{array}{cc} \frac{4\pi i^{n}}{q_{1}^{n+1}u} & \;\mathrm{from}\;\mathrm{in}\text{-}\mathrm{sector}\\ \frac{4\pi i^{n}}{\sqrt{q_{1}^{2n+1}u}} & \;\mathrm{from}\;\mathrm{anywhere}. \end{array}\right. \end{array}$$

Note that the highest degree of *N* must also satisfy ${\left(N+1\right)}^{2}\leq L$, where *L* is the total number of sensors and ${\left(N+1\right)}^{2}$ is the total number of SSH. The series expansion in (Equation 1) can be expressed in matrix form as
(8)$$\begin{array}{ccc} p_{𝓁}\left(k,\Psi_{𝓁}\right) & \approx & g^{T}\left(\Psi_{𝓁}\right)B\left(kr\right)g^{*}\left(\Omega_{s}\right)x\left(k\right), \end{array}$$
where
(9)$$\begin{array}{ccc} g(\Psi_{𝓁}) & = & \mathrm{vec}\left({\left\{{\left\{G_{n}^{m}\left(\Psi_{𝓁}\right)\right\}}_{m=-n}^{n}\right\}}_{n=0}^{N}\right)\in C^{{\left(N+1\right)}^{2}\times 1}, \end{array}$$
(10)$$\begin{array}{ccc} g(\Omega_{s}) & = & \mathrm{vec}\left({\left\{{\left\{G_{n}^{m}\left(\Omega_{s}\right)\right\}}_{m=-n}^{n}\right\}}_{n=0}^{N}\right)\in C^{{\left(N+1\right)}^{2}\times 1}, \end{array}$$
(11)$$\begin{array}{ccc} B(kr) & = & \mathrm{diag}\left(b_{0}\left(kr\right),b_{1}\left(kr\right),b_{1}\left(kr\right),b_{1}\left(kr\right),\cdots ,b_{N}\left(kr\right)\right),\\ & = & \mathrm{diag}\left({\left\{{\left(b_{n}\left(kr\right)\right)}_{\times (2n+1)}\right\}}_{n=0}^{N}\right), \end{array}$$
and the superscript *T* denotes the transpose operator.

For a spherical sector array of *L* microphones, the L×1 received data are
(12)$$\begin{array}{ccc} p(k,\Omega_{s}) & = & G{\left(\Psi \right)}^{T}B\left(kr\right)g^{*}\left(\Omega_{s}\right)x\left(k\right), \end{array}$$
where $G\left(\Psi \right)=\left[g\left(\Psi_{1}\right),g\left(\Psi_{2}\right),\cdots g\left(\Psi_{L}\right)\right]\in C^{{\left(N+1\right)}^{2}\times L}$. This result shows direction–frequency separability, an important feature that we exploit later to develop a focusing matrix for the steered covariance matrix.

### 2.2. First-Order Cardioid Gain Pattern

The response of a first-order cardioid microphone to the incident plane wave is captured by the expression [26,29,30]
(13)$$\begin{array}{ccc} {\tilde{p}}_{𝓁}\left(r,\Omega_{s}\right) & = & \left[\alpha +\left(1-\alpha \right)\cos\left(\gamma_{𝓁}\right)\right]p_{𝓁}\left(r,\Omega_{s}\right), \end{array}$$
where $\gamma_{𝓁}\in \left[0,2\pi \right)$ is the angle between the main axis of the sensor and the direction of arrival of the incident signal, while α∈(0,1) is the shape parameter, also known as the *cardioidicity* index. From (Equation 1), the derivative of incident plane wave with respect to kr is
(14)$$\begin{array}{ccc} -i\frac{𝜕}{𝜕(kr)}p_{𝓁}\left(r,\Omega_{s}\right) & = & \cos\left(\gamma_{𝓁}\right)p_{𝓁}\left(r,\Omega_{s}\right), \end{array}$$
which implies that the second part of (Equation 13) can be obtained via partial derivative of the incident pressure wave. Thus, the collection of outputs of *L* cardioid sensors provides the following:(15)$$\begin{array}{ccc} \tilde{p}\left(k,\Omega_{s}\right) & = & \alpha p\left(k,\Omega_{s}\right)-i\left(1-\alpha \right)\frac{𝜕}{𝜕(kr)}p\left(k,\Omega_{s}\right)\\ & = & G{\left(\Psi \right)}^{T}\left[\alpha B\left(kr\right)-i\left(1-\alpha \right)B^{\prime }\left(kr\right)\right]g^{*}\left(\Omega_{s}\right)x\left(k\right)\\ & = & G{\left(\Psi \right)}^{T}\tilde{B}\left(kr\right)g^{*}\left(\Omega_{s}\right)x\left(k\right) \end{array}$$
where
(16)$$\begin{array}{ccc} \tilde{B}\left(kr\right) & = & \mathrm{diag}\left({\left\{{\left({\tilde{b}}_{n}\left(kr\right)\right)}_{\times (2n+1)}\right\}}_{n=0}^{N}\right), \end{array}$$
(17)$$\begin{array}{ccc} \tilde{b}\left(kr\right) & := & \alpha b_{n}\left(kr\right)-i\left(1-\alpha \right)b_{n}^{\prime }\left(kr\right), \end{array}$$
and
(18)$$\begin{array}{ccc} b_{n}^{\prime }\left(kr\right) & := & \left\{\begin{array}{cc} w_{n}j_{n}^{\prime }\left(kr\right) & \;\mathrm{for}\;\mathrm{open}\;\mathrm{sector}\\ w_{n}\frac{h_{n}^{\left(2\right)}\left(kr\right)}{h_{n}^{{\left(2\right)}^{\prime }}\left(kr\right)}\left[\frac{j_{n}^{\prime }\left(kr\right)h_{n}^{{\left(2\right)}^{\prime \prime }}\left(kr\right)}{h_{n}^{{\left(2\right)}^{\prime }}\left(kr\right)}-j_{n}^{\prime \prime }\left(kr\right)\right] & \;\mathrm{for}\;\mathrm{rigid}\;\mathrm{sector}. \end{array}\right. \end{array}$$
The first derivative of the spherical Bessel function of the first kind with respect to its argument can be expressed as [39]
(19)$$\begin{array}{ccc} j_{n}^{\prime }\left(kr\right) & = & \frac{n}{kr}j_{n}\left(kr\right)-j_{n+1}\left(kr\right), \end{array}$$
with its second derivative as
(20)$$\begin{array}{ccc} j_{n}^{\prime \prime }\left(kr\right) & = & \frac{n(n-1)}{{\left(kr\right)}^{2}}j_{n}\left(kr\right)-\frac{2n+1}{kr}j_{n+1}\left(kr\right)+j_{n+2}\left(kr\right). \end{array}$$
The expressions in (Equation 19) and (Equation 20) also apply for the spherical Hankel function.

Comparing (Equation 15) to (Equation 1), we can deduce the spherical-sector Fourier transform of the pressure field using the cardioid microphone, provided as follows:(21)$$\begin{array}{ccc} p_{n}^{m}\left(kr\right) & = & {\tilde{b}}_{n}\left(kr\right){\left[G_{n}^{m}\left(\Omega_{s}\right)\right]}^{*}. \end{array}$$

### 2.3. Modal Strength of Spherical Sector Array

On comparing the decomposition of the array of omnidirectional sensor and the cardioid sensor, it becomes clear that the directivity of the cardioid sensor affects only the modal strength matrix of the array. In this section, we analyze the modal strength of the open spherical sector array in order to understand how it solves the problem of spherical Bessel null elimination by cardioid microphones. Towards this end, the modal strength of the open sphere $|{\tilde{b}}_{n}\left(kr\right)|$ is expanded as follows:(22)$$\begin{array}{ccc} |{\tilde{b}}_{n}\left(kr\right)| & = & |w_{n}\left|\right|\left(\alpha j_{n}\left(kr\right)-i\left(1-\alpha \right)j_{n}^{\prime }\right)\left(kr\right))|\\ & = & |w_{n}|\left|\alpha j_{n}\left(kr\right)-i\left(1-\alpha \right)\left[\frac{n}{kr}j_{n}\left(kr\right)-j_{n+1}\left(kr\right)\right]\right|. \end{array}$$

Generally, as *n* increases, the amplitude of the spherical Bessel function decreases at small kr; however, the $|w_{n}|$ term decreases as *n* increases for the spherical sector, adding to the clear amplitude difference across *n*. It can be seen that as the value of α decreases, the mode strength decreases with increase in mode kr for order n=0; however, for orders greater than 0, the modal strength is higher for lower values of α. This reduction in zeroth-order mode strength for lower α values is compensated by the elimination of the spherical Bessel function null effect (see Figure 2). This implies that such an array of cardioid sensors has nonzero values for all modes (i.e., frequencies), a problem that affects the array of omnidirectional sensors (see Figure 2c).

The use of cardioid sensors in rigid spherical sector arrays offers some mode strength improvement at very low frequencies, as α decreases for nonzero orders. However, the use of a cardioid with gain pattern containing at least one null introduces ripples in the modal strength of the rigid hemisphere (see Figure 3), making such cardioids less desirable for rigid spherical sectors.

### 2.4. Spatial Resolution of the Spherical Sector Array

Spatial resolution refers to an array’s ability to distinguish between sound sources coming from different directions. High spatial resolution means that the array can differentiate between sources that are close together spatially. In this section, we study the spatial resolution of spherical sector microphone arrays. The sound field directivity function describes how the amplitude or intensity of a sound field varies with direction from a source. We establish the spatial resolution of the array as the main lobe width of the sound field directivity function, as this provides the spatial extent occupied by an incident plane wave. From (Equation 21), the spherical-sector Fourier transform of an incident plane wave from $\Omega =(\theta_{s},\phi_{s})$ is written as follows [40]:(23)$$\begin{array}{ccc} p_{n,s}^{m}\left(kr\right) & = & {\tilde{b}}_{n}\left(kr\right){\left[G_{n}^{m}\left(\theta_{s},\phi_{s}\right)\right]}^{*} \end{array}$$
where all variables are as previously defined. For an infinite number of incident sources arriving from every direction of the spherical sector, we write (Equation 23)
(24)$$\begin{array}{ccc} p_{n}^{m}\left(kr\right) & = & \int_{\Omega_{s}\in S^{2}}w\left(kr,\theta_{s},\phi_{s}\right)p_{n,s}^{m}\left(kr\right)d\Omega_{s}\\ & = & {\tilde{b}}_{n}\left(kr\right)\int_{\Omega_{s}\in S^{2}}w\left(kr,\theta_{s},\phi_{s}\right){\left[G_{n}^{m}\left(\theta_{s},\phi_{s}\right)\right]}^{*}d\Omega_{s}\\ & = & {\tilde{b}}_{n}\left(kr\right)w_{n}^{m}\left(kr\right), \end{array}$$
as the spherical sector Fourier transform. The above integral is over the same spherical sector for which the spherical sector harmonic basis is defined; here, $w(kr,\theta_{s},\phi_{s})$ is the sound field directivity function and $w_{n}^{m}\left(kr\right)$ is its spherical-sector Fourier transform. This implies that
(25)$$\begin{array}{ccc} w_{n}^{m}\left(kr\right) & = & \frac{1}{{\tilde{b}}_{n}\left(kr\right)}p_{n}^{m}\left(kr\right), \end{array}$$
Which we can use to obtain the nature of the sound field directivity function in spatial domain for a plane wave with single-unit amplitude.

Practically, we can only measure the spherical-sector Fourier coefficients of the pressure field up to a limited order *N*. This is due to the limitation imposed by the number of sensors. The directivity of the sound fields in the spatial domain (inverse spherical-sector Fourier transform of $w_{n}^{m}\left(kr\right)$) can only be resolved with finite order *N* for one source. Using (Equation 23) and (Equation 25), we have
(26)$$\begin{array}{ccc} w_{N}\left(\theta ,\phi \right) & = & \sum_{n=0}^{N}\sum_{m=-n}^{n}w_{n}^{m}\left(kr\right)G_{n}^{m}\left(\theta ,\phi \right)\\ & = & \sum_{n=0}^{N}\sum_{m=-n}^{n}\frac{1}{{\tilde{b}}_{n}\left(kr\right)}p_{n}^{m}\left(kr\right)G_{n}^{m}\left(\theta ,\phi \right)\\ & = & \sum_{n=0}^{N}\sum_{m=-n}^{n}{\left[G_{n}^{m}\left(\theta_{s},\phi_{s}\right)\right]}^{*}G_{n}^{m}\left(\theta ,\phi \right). \end{array}$$
Applying the spherical sector harmonics addition theorem [23], we can write (Equation 26) as
(27)$$\begin{array}{ccc} w_{N}\left(\theta ,\phi \right) & = & \sum_{n=0}^{N}\frac{\left(2n+1\right)q_{1}u}{4\pi }P_{n}\left(q_{1}\cos\left(\Theta \right)+q_{2}\right), \end{array}$$
where Θ is the angle between the source direction $\left(\theta_{s},\phi_{s}\right)$ and any direction (θ,ϕ) and is provided as $\cos\left(\Theta \right)=\cos\theta \cos\theta_{s}+\cos\left(\phi -\phi_{s}\right)\sin\theta \sin\theta_{s}$. Applying the Christofell summation formula to (Equation 27), the directivity function of the incident pressure field is
(28)$$\begin{array}{ccc} w_{N}\left(\Theta \right) & = & \frac{\left(N+1\right)q_{1}u}{4\pi }\frac{\left[P_{N+1}\left(q_{1}\cos\Theta +q_{2}\right)-P_{N}\left(q_{1}\cos\Theta +q_{2}\right)\right]}{\left(q_{1}\cos\Theta +q_{2}-1\right)}. \end{array}$$
It can be seen that the directivity function of the plane wave is a function of order *N*, angle Θ, $\left(q_{1},q_{2}\right)$ and *u*, but not a function of the *cardioidicity* index α. Using the ideal beamformer provided in [25] (Equation (Equation 24) for the *ℓ*-th sensor, we have
(29)$$\begin{array}{ccc} w_{𝓁}\left(\theta ,\phi \right) & = & |q_{1}\left||u\right|\sum_{n=0}^{N}\sum_{m=-n}^{n}\frac{1}{{\tilde{b}}_{n}\left(kr\right)}{\left[G_{n}^{m}\left(\theta_{𝓁},\phi_{𝓁}\right)\right]}^{*}G_{n}^{m}\left(\theta ,\phi \right)\\ & = & |q_{1}{|}^{2}{\left|u\right|}^{2}\sum_{n=0}^{N}\frac{1}{{\tilde{b}}_{n}\left(kr\right)}\frac{2n+1}{4\pi }P_{n}\left(q_{1}\cos\Theta +q_{2}\right), \end{array}$$
where Θ is the angle between the sensor direction $\left(\theta_{𝓁},\phi_{𝓁}\right)$ and steering direction (θ,ϕ). The spatial resolution is twice the smallest zero of $w_{𝓁}\left(\theta ,\phi \right)$, which is same as the zeros of (Equation 27), that is, the first zero of $P_{N+1}\left(q_{1}\cos\Theta +q_{2}\right)$. The same result is obtained with the sound field directivity function approach. Notably, this resolution is independent of the cardioid microphone’s α, which only affects the mode strength ${\tilde{b}}_{n}\left(kr\right)$.

For the full sphere (setting u=1, $\left(q_{1},q_{2}\right)=\left(1,0\right)$), the derived directivity function matches that obtained in [40] (Equation (Equation 20)). The plot of the normalized directivity function $w_{N}\left(\Theta \right)$ versus the angle between the source direction and the look direction Θ for various orders *N* is shown in Figure 4.

The main lobe of the directivity function has a width bounded by the first zeros of (Equation 28), $\Theta_{0}$ on both sides. Thus, the spatial resolution of the spherical sector $2\Theta_{0}$ is a function of order *N* and $q_{1}$, $q_{2}$. This main lobe of the sound field directivity function grows narrower as te order *N* increases. This means that more sources can be resolved with higher order *N*, which is directly related the number of sensors.

To illustrate this, we can look at a case of spherical cap; the spherical sector is bounded along the polar angle by $\left[0,\theta_{2}\right]$, where $\theta_{2}\in \left(0,90^{{}^{\circ}}\right]$. To establish a simple equation for the spatial resolution $2\Theta_{0}$ as a function of the order *N* and $q_{1}$ or $\theta_{2}$ of a spherical cap, we can perform a $\Theta_{0}$ versus *N* curve fitting for each $\theta_{2}$ (or $q_{1}$), as follows:(30)$$\begin{array}{ccc} \Theta_{0} & \approx & \frac{\zeta (q_{1})}{N+1},\;\;\;\;\;\;3\leq N\leq 50 \end{array}$$
where $\zeta (q_{1})$ (a function of $q_{1}$ or $\theta_{2}$) is the coefficient of regression. The plot of the spatial resolution $\Theta_{0}$ versus order *N* is shown in Figure 5 for $\theta_{2}=45^{{}^{\circ}}$ and $90^{{}^{\circ}}$. This figure also contains the fitting of the spatial resolution to a rational function of order *N* (excluding N=1 and 2 to achieve a better fit). The spatial resolution values for N<3 can be easily calculated as the smallest root of the Legendre polynomial of orders 2 and 3.

The above fitting was done for $\theta_{2}\in \left[10^{{}^{\circ}},90^{{}^{\circ}}\right]$ in steps of $5^{{}^{\circ}}$, and the values of $\zeta (q_{1})$ were collected for each $\theta_{2}$. An R-squared value of 1 was recorded for all cases, and the root mean square error was bounded by [0.0028,0.0130] degrees. The relationship between $\zeta (q_{1})$ and $q_{1}$ was fitted as follows:(31)$$\begin{array}{ccc} \zeta (q_{1}) & \approx & \frac{219.41}{\sqrt{q_{1}}} \end{array}$$
with a root mean square error of 0.013 degrees and an R-squared value of 1. Combining the fittings in (Equation 30) and (Equation 31), we obtain the final approximation of the spatial resolution ($2\Theta_{0}$) as follows:(32)$$\begin{array}{ccc} 2\Theta_{0} & \approx & \frac{438.82}{\sqrt{q_{1}}\left(N+1\right)},\;\;\;N>2 \end{array}$$
and in terms of the spherical cap limiting the polar angle $\theta_{2}$ as
(33)$$\begin{array}{ccc} 2\Theta_{0} & \approx & \frac{3.5926\;\theta_{2}}{\left(N+1\right)},\;\;\;N>2. \end{array}$$

The values of the spatial resolution are shown in degrees in Table 1 for spherical caps of $\theta_{2}=90^{{}^{\circ}}$ (hemisphere), $\theta_{2}=60^{{}^{\circ}}$, and $45^{{}^{\circ}}$. The spatial resolution is reduced with the reduction in $\theta_{2}$; however, this has to be considered in the context of the spherical cap’s angle limits. For all spherical caps, the ratio of the polar angle limit and the spatial resolution is constant, that is, normalizing the spatial resolution by $\theta_{2}$ provides a spatial resolution that depends only on *N*. However, the reduction in the spherical cap’s polar angle range $\theta_{2}$ means that more sensors are distributed around the azimuth. Hence, for the same order *N*, the smaller the spherical cap, the smaller the spatial resolution.

In the next section, we develop the array manifold interpolation method for the spherical sector array of cardioid microphone to decorrelate the covariance matrix of coherent wideband sources.

## 3. Array Manifold Interpolation for Coherent Wideband Sources

### 3.1. Received Signal Model

We consider *M* wideband sources incident on a spherical sector array of radius *r* comprising *L* first-order cardioid microphones. It is assumed that the bandwidths of the sources overlap within the frequency range $\left[f_{\min},f_{\max}\right]$, corresponding to wavenumber range $\left[k_{\min},k_{\max}\right]$. Thus, the received data are modeled for the *j*-th frequency bin as
(34)$$\begin{array}{ccc} z(k_{j}) & = & A\left(k_{j},\Omega \right)x\left(k_{j}\right)+v\left(k_{j}\right),\;\;j=1,2,\cdots ,J \end{array}$$
where *J* denotes the total number of selected frequency bins, $A(k_{j},\Omega_{s}):=$$\left[\tilde{a}\left(k_{j},\Omega_{1}\right)\;\tilde{a}\left(k_{j},\Omega_{2}\right)\;\cdots \;\tilde{a}\left(k_{j},\Omega_{M}\right)\right]$ is the steering matrix (with $\tilde{a}=\left[\alpha +\left(1-\alpha \right)\right]e^{ik_{s}^{T}r_{𝓁}}$, also expressed as $A\left(k_{j},\Omega \right):=G{\left(\Psi \right)}^{T}\tilde{B}\left(k_{j}r\right)G\left(\Omega \right)$ in spherical sector harmonics), and $\Omega =[\left(\theta_{1},\phi_{1}\right),\left(\theta_{2},\phi_{2}\right),\cdots \left(\theta_{M},\phi_{M}\right)]$ is the collection of the directions of the *M* incident signals. Furthermore, $x\left(k_{j}\right)=[x_{1}\left(k_{j}\right)\;x_{2}\left(k_{j}\right)\;$$\cdots \;x_{M}\left(k_{j}\right){]}^{T}$ is the magnitude of the incident signals at $k_{j}$ and $v(k_{j})$ is the spatiotemporally uncorrelated additive white Gaussian noise, which is also uncorrelated with the signal.

It is assumed that at least two sources are coherent. Given adequate samples, the covariance matrix of the received data is calculated as $R\left(k_{j}\right)=\frac{1}{Q}\sum_{q=1}^{Q}z^{\left(q\right)}\left(k_{j}\right)z^{\left(q\right)}{\left(k_{j}\right)}^{H}$, for the *j*-th frequency bin, where *Q* is the total number of independent observations, $z^{\left(q\right)}\left(k_{j}\right)$ represents the received data for the *q*-th observation, and the superscript ^*H*^ denotes the conjugate transpose. The presence of coherent sources degrades the rank of $R(k_{j})$, giving rise to a singular covariance matrix. This is an issue for beamforming methods that involve inversion of the covriance matrix or for subspace methods that depend on the eigenvalues of the covariance matrix. In the next section, the array manifold interpolation method is proposed for this array to reduce the effect of the correlated sources on direction of arrival estimation.

### 3.2. Array Manifold Interpolation

The performance of narrowband beamforming algorithms significantly deteriorates in the presence of coherent sources. This degradation is particularly evident for algorithms such as Minimum Variance Distortionless Response (MVDR), which rely on the inversion of the covariance matrix [19], as well as for subspace methods that require decomposition of the covariance matrix. Various techniques have been developed to address this issue by decorrelating coherent sources, including spatial smoothing [41] and frequency smoothing [36,42,43,44].

Frequency smoothing involves the coherent summation of the covariance matrices across individual frequency bins. This method has proven to be effective for decorrelating wideband coherent sources. Notably, the focusing matrix proposed by Wang et al. [42] requires an initial estimate of the direction of arrival, in contrast to other methods proposed in [36,43,44] which do not require prior knowledge of the source directions. The methods proposed in [44] require spherical harmonics transformation of the received signal before obtaining the covariance matrix, an unnecessary step that offers no significant advantage and requires extra signal processing to whiten the additive noise in the spherical harmonic domain.

The array manifold interpolation method introduced in this section, similar to the frequency smoothing approach in [36,44], leverages the frequency–direction separability inherent in the spherical harmonics representation of the received data, as demonstrated in (Equation 12).

With the aim of decorrelating coherent sources, the focusing matrix $T(k_{j})$ for various frequency bins j=1,2,⋯,J is such that
(35)$$\begin{array}{ccc} T\left(k_{j}\right)A\left(k_{j},\Omega \right) & = & A(k_{0},\Omega )\\ T\left(k_{j}\right)G{\left(\Psi \right)}^{T}\tilde{B}\left(k_{j}r\right)G\left(\Omega \right) & = & G{\left(\Psi \right)}^{T}\tilde{B}\left(k_{0}r\right)G\left(\Omega \right). \end{array}$$
Eliminating G(Ω) on both sides and multiplying both sides by the inverse of $\tilde{B}\left(k_{j}r\right)$, we have
$$\begin{array}{ccc} T\left(k_{j}\right)G^{T}\left(\Psi \right) & = & G{\left(\Psi \right)}^{T}\tilde{B}\left(k_{0}r\right)\tilde{B}{\left(k_{j}r\right)}^{-1}, \end{array}$$
implying that
(36)$$\begin{array}{ccc} T(k_{j}) & = & G{\left(\Psi \right)}^{T}\tilde{B}\left(k_{0}r\right)\tilde{B}{\left(k_{j}r\right)}^{-1}G{\left(\Psi \right)}^{\#},\;\;\in C^{L\times L}, \end{array}$$
where $G{\left(\Psi \right)}^{\#}:=G\left(\Psi \right){\left(G{\left(\Psi \right)}^{T}G\left(\Psi \right)\right)}^{-1}$. If the number of sensors *L* equals the total number of spherical harmonics ${\left(N+1\right)}^{2}$, then the general inverse operation becomes a matrix inversion operation. Finally, the wideband steered covariance matrix is calculated as [36]
(37)$$\begin{array}{ccc} \tilde{R} & = & \sum_{j=1}^{J}T\left(k_{j}\right)R\left(k_{j}\right)T{\left(k_{j}\right)}^{H}. \end{array}$$
The steered covariance matrix is then used to estimate the direction of arrival using subspace methods such as the MUSIC algorithm. A summary of this method is described in Algorithm 1.
**Algorithm 1** Summary of DOA estimation steps1:**Input:** Received multichannel signal data z(t) across *L* channels2:**Output:** Estimated Directions of Arrival (DOA) $\hat{\Omega }=\left(\hat{\theta_{s}},\;\hat{\phi_{s}}\right)$, s=1,2,⋯,M sources.3:**Step 1: Fourier Transform of Received Data**4:**for** each channel 𝓁=1 to *L* **do**5:    Perform Fourier Transform on time-domain signal to obtain frequency-domain representation $Z_{𝓁}\left(f\right)$6:**end for**7:**Step 2: Frequency Bin Selection, Covariance and Focusing Matrices Generation**8:Select *J* frequency bins within the signal bandwidth9:**for** each selected frequency bin (i.e., wave number $k_{j}$) j=1 to *J* **do**10:    Compute covariance matrix $R(k_{j})$ for each bin and,11:    Compute focusing matrix $T(k_{j})$ using (Equation 36) for each bin12:**end for**13:**Step 4: Generate the Steered Covariance Matrix**14:Compute the steered covariance matrix $\tilde{R}$ using (Equation 37)15:**Step 5: DOA Estimation using MUSIC algorithm**16:**for** each possible direction (θ,ϕ) **do**17:    Calculate MUSIC pseudo-spectrum18:**end for**19:Identify peaks in the MUSIC spectrum, which correspond to estimated DOA angles $\hat{\Omega }=\left(\hat{\theta_{s}},\hat{\phi_{s}}\right)$.

## 4. Direction-of-Arrival Performance Comparison

In this section, we evaluate the efficacy of the proposed method by testing its performance in terms of DOA accuracy, particularly for coherent sources. The assessment is carried out through a series of simulations using corresponding sensor locations of the Eigenmike array [45], a spherical microphone array with near-uniformly sampling. The test scenario is designed to focus on a hemisphere, as shown in Figure 1.

Through this rigorous testing, we aim to demonstrate the efficacy of the proposed method for coherent sources and to compare the performance of cardioid and omnidirectional microphone arrays under this condition.

For our simulations, we used the hemisphere bounded by $\theta \in [0,90^{{}^{\circ}}]$ and $\phi \in [0,360^{{}^{\circ}})$. The hemisphere is sampled near-uniformly according to the sensor positions on the Eigenmike em32 microphone array of radius 4.2 cm, which corresponds to 20 sensors distributed near-uniformly over the hemisphere. The wideband sources are simulated as coherent zero-mean band-limited Gaussian processes incident from four directions $\left(\theta_{s},\phi_{s}\right)=\{\left(60^{{}^{\circ}},60^{{}^{\circ}}\right)$, $\left(30^{{}^{\circ}},150^{{}^{\circ}}\right)$, $\left(80^{{}^{\circ}},220^{{}^{\circ}}\right)$, $\left(30^{{}^{\circ}},300^{{}^{\circ}})\right\}$ and occupying frequency range f∈[1.2,1.8]kHz. For these four sources, the complex scalars uesd to model the coherence are $\left(1,0.75e^{j1.2^{{}^{\circ}}},0.77e^{j0.03^{{}^{\circ}}},0.76e^{j0.4^{{}^{\circ}}}\right)$. The step-by-step procedure for generating the received data is provided in Algorithm 2.

**Algorithm 2** Simulating received signal

1:**Input**: Number of samples *K*, signal bandwidth $f_{\mathrm{stop}}=\left(1.2,1.8\right)$ kHz and $f_{\mathrm{pass}}=\left(1.275,1.725\right)$ kHz, sampling frequency $f_{s}$, number of sources *M*, source directions $\Omega =\{\left(\theta_{1},\phi_{1}\right),\left(\theta_{2},\phi_{2}\right),\cdots ,\left(\theta_{M},\phi_{M}\right)\}$, complex scalars, and signal-to-noise ratio (SNR).2:**Output**: The L×K received signal comprising array response to coherent sources arriving from distinct directions and AWGN across each channels.3:
**1. Generate Complex Gaussian Random Variable**
4:Generate *K* samples of complex Gaussian random samples $x∼N_{c}\left(0,\sigma_{s}^{2}\right)$.5:Store *x* as the initial signal representing a source.6:
**2. Filter the Gaussian Signal**
7:Design a Kaiser window FIR bandbass filter using $f_{\mathrm{stop}}$ and $f_{\mathrm{pass}}$ constraints8:Apply the designed bandpass filter to *x* to obtain $x_{\mathrm{filtered}}$, a band-limited version of the original Gaussian signal.9:
**3. Array Response to the *M* sources**
10:Multiply $x_{\mathrm{filtered}}$ by its corresponding steering vector $\tilde{a}\left(k,\Omega_{1}\right)$ to obtain the array’s response to source 111:**for** each additional source s=2,⋯,M **do**12:    Model $x_{s}$ as product of complex scalar and $x_{\mathrm{filtered}}$: $x_{s}=\beta_{s}x_{\mathrm{filtered}}e^{j\psi_{s}}$, where $\beta_{s}$ is the scaling factor and $\psi_{s}$ is the phase shift for source *s*.13:    Multiply $x_{s}$ by the steering vector $\tilde{a}\left(k,\Omega_{s}\right)$ associated with its direction of arrival $\Omega_{s}=\left(\theta_{s},\phi_{s}\right)$ to obtain the array response to the $x_{s}$,14:
**end for**
15:Sum all array responses to the *M* sources to produce the resultant array response16:
**4. Simulate Additive White Gaussian Noise (AWGN)**
17:**for** each channel 𝓁=1,⋯,L **do**18:    Estimate the signal power $P_{\mathrm{signal}}$ of the *ℓ*-th channel.19:    Generate zero-mean complex Gaussian noise $v_{𝓁}∼N_{c}\left(0,\sigma_{v}^{2}\right)$ with variance based on the target SNR: $\sigma_{v}^{2}=\frac{P_{\mathrm{signal}}}{\mathrm{SNR}}$20:    Add $v_{𝓁}$ to the corresponding channel signal.21:
**end for**
22:
**8. Output the Received Signal**
23:The final received signal is the sum of the array response and AWGN for each channel.



For processing, we used the center frequency of $f_{0}=1.5\;\mathrm{kHz}$ and 20 frequency bins. Plots of the sample beampattern versus the polar and azimuthal angles for various values of α are shown in Figure 6 for an open hemisphere. For the open hemisphere, the spatial spectrum degrades as the value of α increases towards 1. The open hemispherical array of omnidrectional microphones produced the least smooth spatial spectrum (see Figure 6d). This can be attributed to the spherical Bessel function null for some frequencies in the modal strength of this array, as discussed in Section 2.3.

The rigid hemisphere shows a different trend. The plots of the sample beampattern versus the polar and azimuthal angle for various values of α are shown in Figure 7 for a rigid hemisphere. For the rigid hemispherical array of cardioid sensors, the performance in terms of clear peak resolution degrades with the presence of nulls in the cardioid sensor’s gain pattern. This can be noted in Figure 7a (hypercardioid with two nulls) and Figure 7b (subcardioid with no null). It can be seen that the rigid sphere of subcardioids outperforms the omnidirectional microphones. This is attributed to the higher modal strength of the subcardioid at lower frequencies for spherical sector harmonics of n>0 (see Figure 2).

### Comparison with Different Signal-to-Noise Ratios

In this section, the performance of the arrays is studied for different signal-to-noise ratios (SNRs). To vary the SNR, white Gaussian noise is added to the simulated signal such that the SNR is varied from 0dB to 40dB according on Algorithm 2. Two sources with the same previously defined spectral characteristics arrive from $\left(\theta ,\phi \right)=\left(60^{{}^{\circ}},60^{{}^{\circ}}\right)$ and $\left(30^{{}^{\circ}},150^{{}^{\circ}}\right)$. We define the cumulative mean error (CME), an estimation error based on the inner angle between a unit vector in the true source direction $\left(\theta_{s},\phi_{s}\right)$ and a unit vector in the estimated source direction $\left({\hat{\theta }}_{s},{\hat{\phi }}_{s}\right)$:(38)$$\begin{array}{ccc} {\mathrm{CME}}_{\theta ,\phi } & := & \frac{1}{MT_{r}}\sum_{t_{r}=1}^{T_{r}}\sum_{s=1}^{M}\cos^{-1}\left({\hat{e}}_{s(t_{r})}\left(\hat{\theta },\hat{\phi }\right)\right) \end{array}$$
where ${\hat{e}}_{s(t_{r})}\left(\hat{\theta },\hat{\phi }\right):=\cos\left(\phi_{s}-{\hat{\phi }}_{s(t_{r})}\right)\sin\left(\theta_{s}\right)\sin\left({\hat{\theta }}_{s(t_{r})}\right)+\cos\left(\theta_{s}\right)\cos\left({\hat{\theta }}_{s(t_{r})}\right)$, $T_{r}$ is the total number of Monte Carlo trials and *M* is the total number of sources. By defining the error as the inner angle of the two vectors, the need to normalize the polar and azimuth angle is avoided, as we demonstrate later. This provides general insight into the performance of the arrays. The cumulative mean error performance versus the signal-to-noise ratio is studied for different α. For each SNR value, the CME is calculated for 1000 iterations.

Plots of the cumulative mean error ${\mathrm{CME}}_{\theta ,\phi }$ versus the signal-to-noise ratio for various α are shown in Figure 8 for the open hemisphere (Figure 8a) and rigid hemisphere (Figure 8b). For the open hemisphere, the CME value increases with the value of α for all considered SNR values. This is in consonance with the analysis in Section 2.3. For the rigid hemisphere, the array of cardioid microphones outperformed the array of omnidirectional microphones. This performance varies with direction of arrival, as the performance of the omnidirectional microphone array tends to be consistent across different directions of arrival.

It is also necessary to study the performance of these arrays in terms of the root mean square error (RMSE) when estimating the polar and azimuthal angles. This can provide information on the directional angles for which one array performs better than the other, representing a key insight into their respective polar versus azimuthal angle resolution. For the open hemisphere, the plot of the root mean square estimation error versus the signal-to-noise ratio is shown in Figure 9. The standard cardioid microphone array performs better in terms of polar angle RMSE at higher SNRs, while the hypercardioid array shows the worst case at higher SNRs (see Figure 9a). However, for the azimuth angle RMSE, the hypercardioid array provides consistently superior performance, followed by the standard cardioid, subcardioid, and finally the omnidirectional microphone array.

For the rigid hemisphere (see Figure 10), the RMSE of the polar angle increases as the value of α decreases (see Figure 10a). The reverse is the case for the RMSE of the azimuthal angle, where the RMSE increases as α increases. Thus, we can conclude that both open and rigid hemispherical microphone arrays of cardioid microphones exhibit higher azimuthal angle estimation accuracy compared to omnidirectional microphone arrays, while the reverse is not always the case for polar angle estimation accuracy.

Thus far, our analysis has evaluated array performance across various SNR levels for specific directions of arrival (DoA). Next, we define a set of DoA values for Source 1 characterized by a polar angle θ ranging from $10^{{}^{\circ}}$ to $90^{{}^{\circ}}$ in increments of $10^{{}^{\circ}}$ and an azimuth angle ϕ ranging from $0^{{}^{\circ}}$ to $360^{{}^{\circ}}$ (exclusive) in steps of $60^{{}^{\circ}}$. This results in 9×6=54 unique directions. The DoA for Source 2 (a coherent source) is modeled as random directions within the same polar and azimuth angle ranges with a complex scalar $0.78e^{j1.15^{{}^{\circ}}}$ to model the coherence. For each DoA pair (for Source 1 and Source 2), 200 trials were conducted at a given SNR. The root mean square error (RMSE) of the polar and azimuthal angles were calculated for each pair.

In total, we obtained 54 RMSE values corresponding to all DoA pairs for a given cardioid type. The mean, minimum, and maximum RMSE values of these 54 direction pairs are computed and visualized in Figure 11 for an SNR of 10 dB and in Figure 12 for an SNR of 50 dB.

It can be observed that the average RMSE of the polar angle is lower than the average RMSE of the azimuthal angle for all cases. This implies that the near-uniformly sampled hemispherical array has relatively higher polar angle resolution. On an open hemisphere, for anSNR of 10 dB (i.e., Figure 11a), the average RMSE of both the polar and azimuthal angles increases with α. This trend is similarly observed for an SNR of 50 dB (see Figure 12a). On a rigid hemisphere, the disparity in performance across the cardioid microphones is small for both SNRs (see Figure 11b and Figure 12b). Again, it can be noticed that the standard cardioid and subcardioid have the best performance for polar angle estimation, while the hypercardioid has the best performance for azimuthal angle estimation.

## 5. Conclusions

In this paper, we have developed and studied a spherical sector array of first-order cardioid microphones. The proposed model is based on the spherical sector harmonic (SSH) basis function, which extends the benefits of spherical harmonics for spherical sector array processing. A modal strength analysis shows that the use of cardioid microphones in open spherical sectors offers higher modal strength for nonzero orders while eliminating the nulls of the spherical Bessel function. Our study of the spatial resolution of the spherical cap has revealed that the spatial resolution of the spherical cap array is independent of the constituent microphone gain pattern, but depends on the maximum order of the array and the limiting polar angle of the spherical cap.

In a performance comparison of direction-of-arrival estimation, we used the array manifold interpolation method to derive a steering matrix for the steered covariance matrix of coherent wideband sources. In order to understand how directional microphones perform for spherical sector arrays, comparisons were conducted between the performance of a spherical sector array of first-order cardioid microphones and an omnidirectional microphone array for wideband sources. In consonance with the theoretical analysis, our simulation results revealed that the open hemispherical array of cardioid microphones generally outperformed the array of omnidirectional microphones. Furthermore, this performance advantage increased with the directivity of the cardioid microphone array, with the hypercardioid demonstrating the best performance. For the rigid hemispherical array, cardioid microphones again outperformed omnidirectional microphones; however, cardioid microphones with no nulls performed better than those with nulls in terms of the gain pattern, indicating that subcardioid microphones are the best choice for rigid hemispherical arrays.

## Figures and Tables

**Figure 1 sensors-24-07572-f001:**
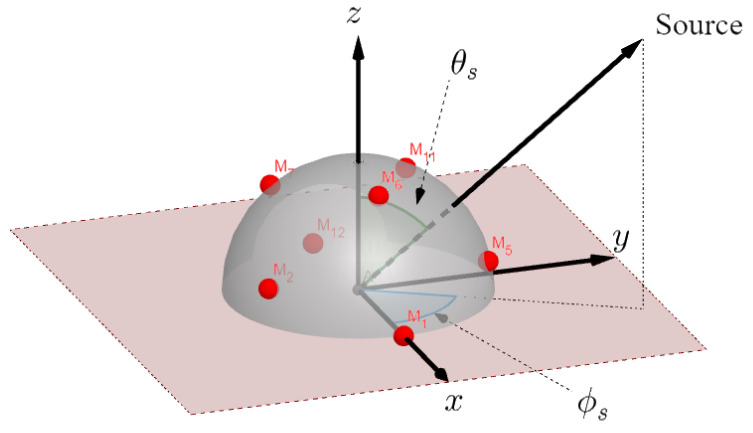
Hemispherical microphone array showing positions of nearly-uniformly sampled microphone locations.

**Figure 2 sensors-24-07572-f002:**
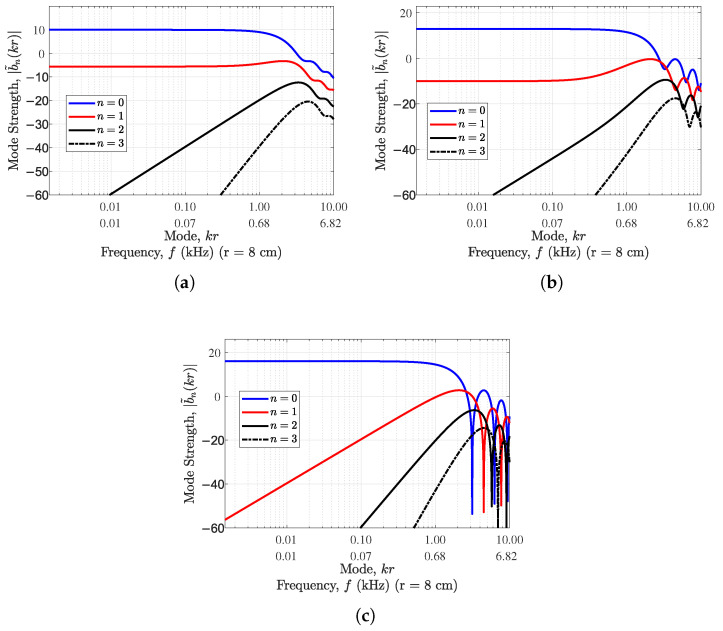
Mode strength variation with mode kr for different degrees of *n* and (**a**) α=0.5 (standard cardioid), (**b**) α=0.7 (subcardioid), and (**c**) α=1 (omnidirectional) for a open hemisphere of radius 8cm.

**Figure 3 sensors-24-07572-f003:**
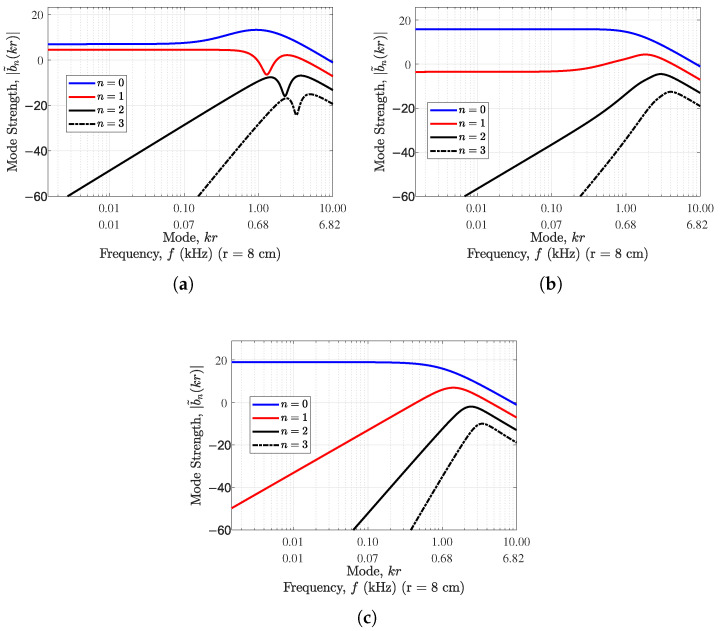
Mode strength variation with mode kr for different degrees of *n* and (**a**) α=0.25 (hypercardioid), (**b**) α=0.7 (subcardioid), and (**c**) α=1 (omnidirectional) for a rigid hemisphere of radius 8cm.

**Figure 4 sensors-24-07572-f004:**
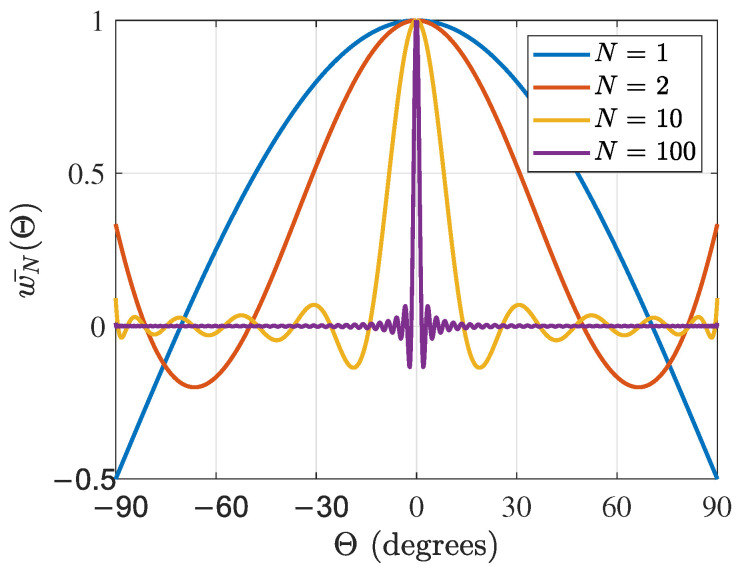
Plot of the normalized directivity function $w_{N}\left(\Theta \right)$ versus the angle between the source direction and look direction Θ using a hemisphere for various orders *N*.

**Figure 5 sensors-24-07572-f005:**
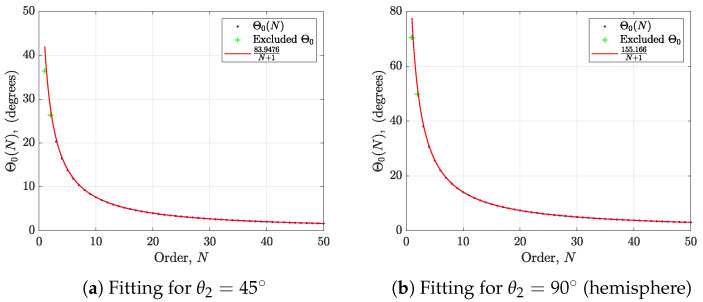
Plot of the spatial resolution $\Theta_{0}$ versus order *N* and fitting to a rational function $\frac{a}{N+1}$ for different values of $\theta_{2}$.

**Figure 6 sensors-24-07572-f006:**
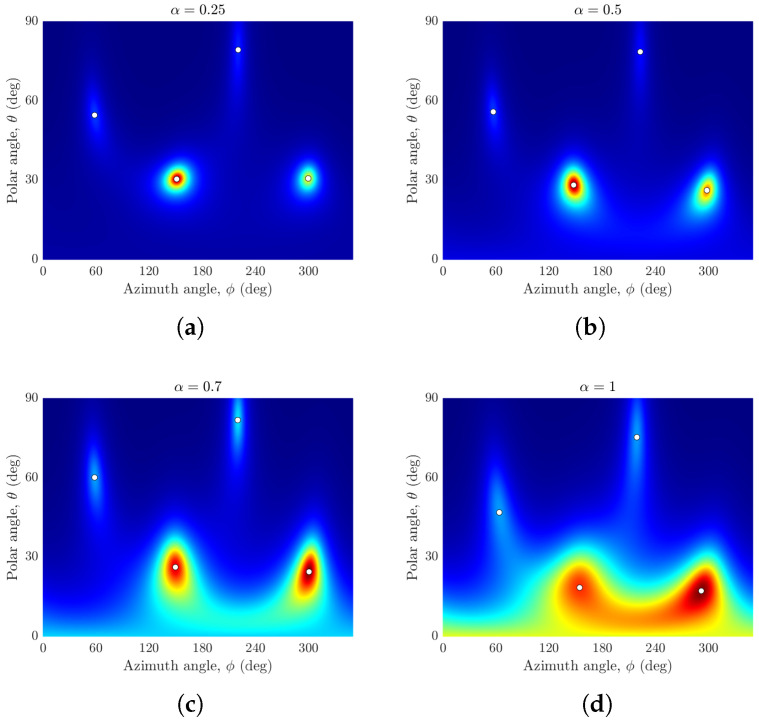
Plots of the beampattern magnitude versus the polar angle θ and azimuthal angle ϕ for four incident coherent sources from $\left(\theta_{s},\phi_{s}\right)=\left\{\left(60^{{}^{\circ}},60^{{}^{\circ}}\right),\left(30^{{}^{\circ}},150^{{}^{\circ}}\right),\left(80^{{}^{\circ}},220^{{}^{\circ}}\right),\left(30^{{}^{\circ}},300^{{}^{\circ}}\right)\right\}$ using an open hemisphere of (**a**) α=0.25 (hypercardioid), (**b**) α=0.5 (standard cardioid), (**c**) α=0.7 (subcardioid), and (**d**) α=1 (omnidirectional).

**Figure 7 sensors-24-07572-f007:**
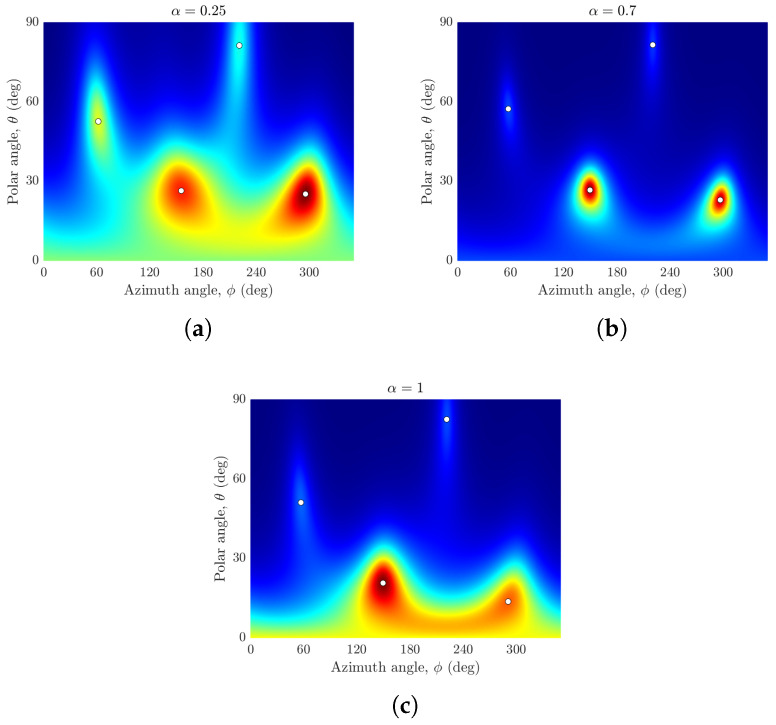
Plots of the beampattern magnitude versus the polar angle θ and azimuthal angle ϕ for four incident coherent sources from $\left(\theta_{s},\phi_{s}\right)=\left\{\left(60^{{}^{\circ}},60^{{}^{\circ}}\right),\left(30^{{}^{\circ}},150^{{}^{\circ}}\right),\left(80^{{}^{\circ}},220^{{}^{\circ}}\right),\left(30^{{}^{\circ}},300^{{}^{\circ}}\right)\right\}$ using a rigid hemisphere of (**a**) α=0.25 (hypercardioid), (**b**) α=0.7 (subcardioid), and (**c**) α=1 (omnidirectional).

**Figure 8 sensors-24-07572-f008:**
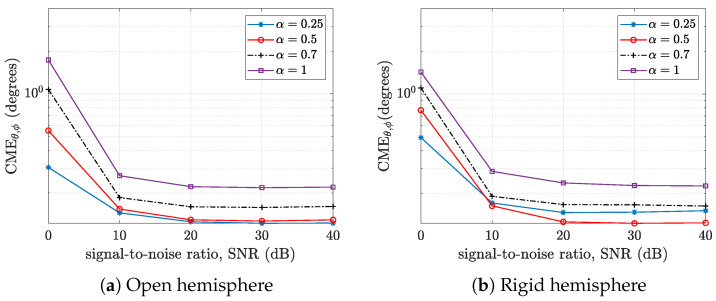
Plots of the cumulative mean error ${\mathrm{CME}}_{\theta ,\phi }$ versus the signal-to-noise ratio (SNR) for various α in (**a**) open hemisphere and (**b**) rigid hemisphere.

**Figure 9 sensors-24-07572-f009:**
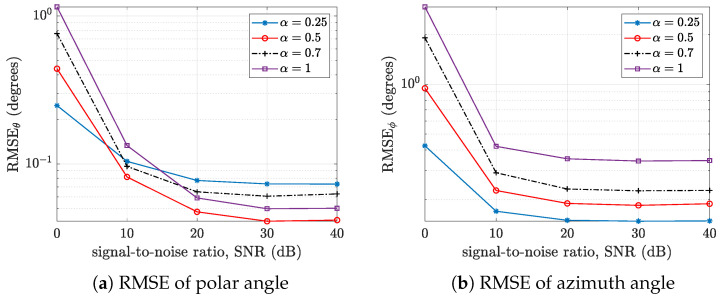
Plots of the root mean square error (RMSE) versus the signal-to-noise ratio (SNR) of an open hemisphere with various α for (**a**) the polar angle θ and (**b**) the azimuthal angle ϕ.

**Figure 10 sensors-24-07572-f010:**
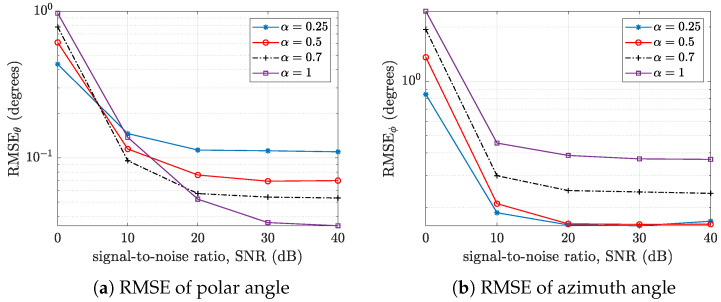
Plots of the root mean square error (RMSE) versus the signal-to-noise ratio (SNR) of a rigid hemisphere with various α: (**a**) the polar angle θ and (**b**) the azimuthal angle ϕ.

**Figure 11 sensors-24-07572-f011:**
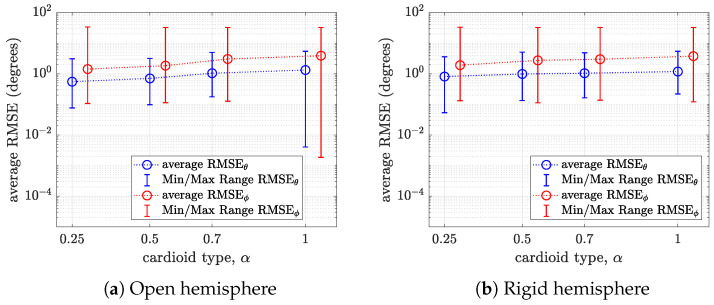
Plots of the average RMSE and minimum and maximum RMSE of the polar and azimuthal angles versus the cardioid type α for (**a**) an open hemisphere and (**b**) a rigid hemisphere for an SNR of 10 dB.

**Figure 12 sensors-24-07572-f012:**
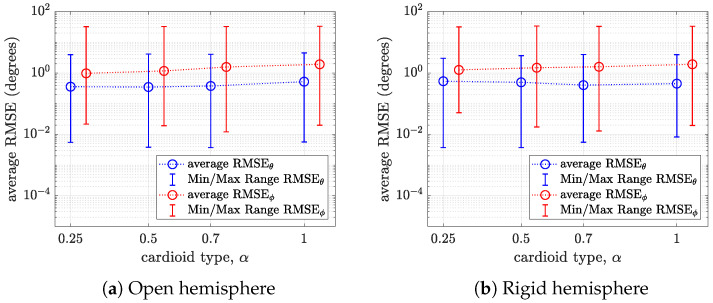
Plots of the average RMSE and minimum and maximum RMSE of the polar and azimuthal angles versus the cardioid type α for (**a**) an open hemisphere and (**b**) a rigid hemisphere for an SNR of 50 dB.

**Table 1 sensors-24-07572-t001:** Spatial resolution $2\Theta_{0}$ (in degrees) for some order *N* and spherical caps ($\theta_{2}$).

*N*	$\theta_{2}=90^{{}^{\circ}}$	$\theta_{2}=60^{{}^{\circ}}$	$\theta_{2}=45^{{}^{\circ}}$
3	80.8	53.9	40.4
4	64.7	43.1	32.3
5	53.9	35.9	26.9
6	46.2	30.8	23.1

## Data Availability

Data are contained within the article.

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
