# Peer review of "Performance Analysis of Cardioid and Omnidirectional Microphones in Spherical Sector Arrays for Coherent Source Localization"

_sensors, 2024, doi:10.3390/s24237572_

Round 1

Reviewer 1 Report

Comments and Suggestions for Authors

The main attention in the article is paid to the influence of the directivity of microphones on the effectiveness of the system of noise sources localization.

 The authors should describe the simulation series in more detail, they conducted to assess the effect of microphone directivity on DOA accuracy, since the description presented is very concise and raises a few questions.

 The authors note that the noise sources are broadband and are modelled as coherent white Gaussian noise. Note, however, that white noise as such cannot be coherent. The concept of coherence refers to two processes as a measure of their connection. For random processes, this relationship is determined by the correlation interval, which tends to zero for broadband white noise. Therefore, it remains unclear how four sources of coherent broadband white noise are simulated in the system, unless they are not the same source.

In addition, simulations in the work were carried out for cases of open and rigid hemispheres. At the same time, the article does not disclose what physical conditions such surfaces correspond to and how they are implemented during simulations.

Obviously, the materials presented in the article would become more understandable if the authors provided a structural diagram of the considered system with an array of directional microphones showing its operation, for example, when simulations.

Reviewer 2 Report

Comments and Suggestions for Authors

This paper investigates the effectiveness of cardioid versus omnidirectional microphone array in spherical sector arrays, with a focus on direction-of-arrival (DOA) estimation and spatial resolution for coherent wideband sources. Overall, the paper is well-written, but it may be challenging for readers who are not specialists in this particular field. I understand that conveying advanced concepts sometimes necessitates omitting foundational details, but I believe that including some figures to visualize the overall ideas and approaches would greatly benefit researchers from related areas. I have read several papers by the first author, but unfortunately, I find this paper to be the least approachable in terms of accessibility. It seems to primarily present final equations and graphs without much supporting context.

Here are my specific recommendations:

1. Generally, omnidirectional microphone array perform well for sound localization, but in your paper, the findings suggest otherwise. I recommend including a few general statements summarizing the main findings. It would also be helpful to include an illustration of what the microphone array look like.

2. The caption for Figure 2 is incorrect.

3. Why do you use radians in Figure 4? All other equations and figures are already using radians.

4. For Figure 8, please clarify whether the values are in degrees or radians.

Thank you to the authors for providing valuable insights into sound localization. I appreciate the effort in exploring the intricacies of microphone arrays and hope future work will expand these findings for a broader audience. More accessible visual aids and explanations could enhance the impact of this research.
